# Lipid Profile, Lipase Bioactivity, and Lipophilic Antioxidant Content in High Pressure Processed Donor Human Milk

**DOI:** 10.3390/nu11091972

**Published:** 2019-08-21

**Authors:** Aleksandra Wesolowska, Joanna Brys, Olga Barbarska, Kamila Strom, Jolanta Szymanska-Majchrzak, Katarzyna Karzel, Emilia Pawlikowska, Monika A. Zielinska, Jadwiga Hamulka, Gabriela Oledzka

**Affiliations:** 1Laboratory of Human Milk and Lactation Research at Regional Human Milk Bank in Holy Family Hospital, Department of Neonatology, Faculty of Health Sciences, Medical University of Warsaw, Zwirki i Wigury Str. 63A, 02-091 Warsaw, Poland; 2Department of Chemistry, Faculty of Food Sciences, Warsaw University of Life Sciences—SGGW, Nowoursynowska St. 166, 02-787 Warsaw, Poland; 3Department of Medical Biology, Faculty of Health Sciences, Medical University of Warsaw, 14/16 Litewska St., 00-575 Warsaw, Poland; 4Department of Biochemistry, Second Faculty of Medicine, Medical University of Warsaw, Banacha 1b Str. 02-093 Warsaw, Poland; 5Faculty of Psychology, Warsaw University, Stawki 5/7, 00-183 Warsaw, Poland; 6High Pressure Physics, Polish Academy of Science, Sokolowska 29, 01-142 Warsaw, Poland; 7Department of Human Nutrition, Faculty of Human Nutrition and Consumer Sciences, Warsaw University of Life Sciences—SGGW, 159 Nowoursynowska St., 02-776 Warsaw, Poland

**Keywords:** donor human milk, high pressure processing, carotenoids, antioxidant capacity, lipids, bile salt stimulated lipase, preterm

## Abstract

Human milk fat plays an essential role as the source of energy and cell function regulator; therefore, the preservation of unique human milk donors’ lipid composition is of fundamental importance. To compare the effects of high pressure processing (HPP) and holder pasteurization on lipidome, human milk was processed at 62.5 °C for 30 min and at five variants of HPP from 450 MPa to 600 MPa, respectively. Lipase activity was estimated with QuantiChrom™ assay. Fatty acid composition was determined with the gas chromatographic technique, and free fatty acids content by titration with 0.1 M KOH. The positional distribution of fatty acid in triacylglycerols was performed. The oxidative induction time was obtained from the pressure differential scanning calorimetry. Carotenoids in human milk were measured by liquid chromatography. Bile salt stimulated lipase was completely eliminated by holder pasteurization, decreased at 600 MPa, and remained intact at 200 + 400 MPa; 450 MPa. The fatty acid composition and structure of human milk fat triacylglycerols were unchanged. The lipids of human milk after holder pasteurization had the lowest content of free fatty acids and the shortest induction time compared with samples after HPP. HPP slightly changed the β-carotene and lycopene levels, whereas the lutein level was decreased by 40.0% up to 60.2%, compared with 15.8% after the holder pasteurization.

## 1. Introduction

Lipids in human milk are not only the main source of energy, but also of bioactive components and regulatory factors, such as vitamins and polyunsaturated fatty acids. Lipids exhibit several functions in the range of biological effects connected with neurodevelopment, immunity, digestion, metabolism regulation, cell membranes communication, and signal transduction [1]. Carotenoids, which are lipid-soluble provitamins, contribute to the anti-oxidative properties of human milk, protecting preterm infants against toxic free radicals [2]. Human milk contains a high bile salt stimulated lipase (BSSL) concentration, which enables the easy digestion of mother’s milk lipids even in the absence of the enzyme in premature newborns. Human milk lipids are valuable, nourishing food with high bioavailability for newborns [3]. Therefore, expressed human milk, including donor milk, should be handled with care to minimize the loss of the unique lipid composition and lipase activity [4]. Administration of donor milk within the hospital setting often requires several preparatory stages, such as pumping, freezing, thawing, and pasteurization by heat treatment (usually holder pasteurization, HoP). The potential losses in human milk lipid content could accumulate when milk passes through all these stages from donor to recipients [5]. Earlier research indicates that heat treatment does not decrease lipid content and does not alter the fat soluble vitamins A, D, and E [6]. However, it was recently shown, using mid-infrared spectroscopy, that lipids are one of the most affected components in pooled pasteurized milk compared with raw milk donated to a milk bank [7]. It was reported earlier that the total availability of lipids in donor milk is affected by the decrease of BSSL and high adhesion to containers surface [8]. In addition, human milk processing may decrease the antioxidant properties of milk, which could have nutritional and clinical implications owing to the particular susceptibility of polyunsaturated fats to peroxidation. 

One of promising preservation methods that allows the unique properties of human milk to be maintained unchanged is high pressure processing (HPP). It is a non-thermal technology that is being increasingly applied in food industries worldwide. This method consists of applying hydrostatic high pressure in short-term treatment to inactivate pathogenic microorganisms and provide nutritionally intact and sensory high-quality products [9].

The aim of our study was to evaluate high pressure processing as a promising technique for the preservation the lipid profile, antioxidant properties, and lipase enzymes activity in donor milk.

## 2. Materials and Methods

### 2.1. Ethical Approval

The Bioethics Committee of Warsaw Medical University has accepted the information about conducting this non-interventional study without reservations (admission number AKBE/59/15). 

### 2.2. Milk Sampling

Milk samples were obtained from donors between the second and sixth week of lactation from Regional Human Milk Bank in Warsaw at Holy Family Hospital. Women were given standard instructions about the best practices for expressing breast milk. Milk samples of approximately 50 mL were collected at home or in the hospital ward using an electric or manual pump, stored in the refrigerator at the temperature of 4 °C, and delivered to the human milk bank within 24 h, while maintaining refrigerated conditions. 

To carry out the study, 6 pooled samples from 3–4 donors were used. Experiments were performed in duplicate and repeated three times with similar results in the following variants: -raw milk—control-temperature processed milk/holder pasteurization (HoP)—comparator-high pressure processed (HPP)—experimental

The processing influence on lipase activity and the change in human milk lipids were evaluated according to these experimental designs. 

Milk samples for all analyses, except for lipase activity, were centrifuged at 4400 rpm for 15 min at 4 °C (Centrifuge 5702R, Eppendorf, Darmstadt, Germany) he lipid monolayer was carefully separated, and both supernatants and lipid monolayer were frozen for later analysis. Lipase activity assay was performed on whole milk samples. 

### 2.3. Human Milk Processing

#### 2.3.1. High Pressure Processing (HPP)

High pressure processing was applied to human milk samples using U 4000/65 apparatus designed and produced by Unipress Equipment at the Institute of High Pressure Physics, Polish Academy of Sciences. The maximum pressure available in the apparatus was 600 MPa and the treatment chamber had a distilled water and polypropylene glycol mixture (1:1), used as pressure-transmitting fluid. The manufacturer designed the working temperature ranges of the apparatus between −10 °C and +80 °C. In our experiments, the test condition temperature was between 19 and 21 °C. Pressure was applied in the following variants: (1) 600 MPa, 10 min; (2) 100 MPa, 10 min, interval 10 min, 600 MPa, 10 min; (3) 200 MPa, 10 min, interval 10 min, 400 MPa, 10 min; (4) 200 MPa, 10 min, interval 10 min, 600 MPa, 10 min; (5) 450 MPa, 15 min. The pressure generation time was 15–25 s and the decompression time was 1–4 s.

#### 2.3.2. Holder Pasteurization (HoP)

Human milk samples were pasteurized according to the routine procedure performed in Regional Human Milk Bank in Warsaw at Holy Family Hospital, using Sterifeed S90 ECO Pasteurise (Medicare Colgate Ltd., Devon, UK, at 62.5 °C for 30 min. 

### 2.4. Lipase Activity

Lipase activity was determined using the lipase assay kit (QuantiChrom Lipase Assay Kit, Bioassay Systems, Hayward, CA, USA) according to the kit instructions. Whole fat human milk samples were brought to room temperature and diluted with distilled water (1:250, *v/v*). The optical density was measured by a Biotek multiple reader at room temperature.

### 2.5. Lipidome Profile

#### 2.5.1. Fat Extraction

The Folch method [10] with further improvement by Boselli [11] was applied to extract fat from the studied samples.

#### 2.5.2. The Oxidative Induction Time

Induction time (IT) was determined by the use of PDSC (pressure differential scanning calorimetry) to evaluate the samples’ oxidative stability. A differential scanning calorimeter (DSC Q20 TA Instruments, New Castle, DE, USA) coupled with a high-pressure cell was used. Fat samples of 3–4 mg were weighed into an aluminium open pan and placed in the sample chamber under oxygen atmosphere with an initial pressure of 1400 kPa. The isothermal temperature for each sample was 120 °C. Obtained diagrams were analyzed using TA Universal Analysis 2000 software (TA Instruments, New Castle, DE, USA). For each sample, the output was automatically recalculated and presented as the amount of energy per one gram. The maximum oxidation induction time was determined based on the maximum rate of heat flow.

#### 2.5.3. Acid Values and Free Fatty Acids Content

The acid value was determined by titration of fat samples with 0.1 M ethanolic potassium hydroxide solution. Free fatty acids (FFA) concentration was calculated based on acid values and the oleic acid molar mass value. Acid values were determined according to ISO method 660:2000.

#### 2.5.4. Fatty Acid Composition

The determination of fatty acid composition was carried out by gas chromatographic (GC) analysis of fatty acid methyl esters. Methyl esters of fatty acids were prepared through transesterification with sodium methoxide according to ISO 5509:2001. A YL6100 GC chromatograph equipped with a flame ionization detector and BPX-70 capillary column of 0.25 mm i.d. x 60 m length and 0.25 μm film thickness was used. The oven temperature was programmed as follows: 60 °C for 5 min; then it was increased by 10 °C/min to 180 °C; from 180 °C to 230 °C, it was increased by 3 °C/min; and then it was kept at 230 °C for 15 min. The temperature of the injector was 225 °C, with a split ratio of 1:50 and the detector temperature of 250 °C. Nitrogen was used as the carrier gas. The results were expressed as relative percentages of each fatty acid, calculated by external normalization of the chromatographic peak area. Fatty acids were identified by comparing the relative retention times of fatty acid methyl ester (FAME) peaks with FAME chemical standard.

#### 2.5.5. Positional Distribution of Fatty Acid in Tag

Positional distribution of fatty acid in sn-2 and sn-1,3 positions in triacylglycerols (TAG) was based on the pancreatic lipase ability to selectively hydrolyze ester bonds in sn-1,3 positions. Briefly, 20 mg of purified pancreatic lipase (porcine pancreatic lipase, crude type II), 1 mL of Tris buffer (pH 8.0), 0.25 mL of bile salts (0.05%), and 0.1 mL of calcium chloride (2.2%) were added to 50 mL centrifuge tubes and vortexed with 0.1 g of fat sample. The mixture was incubated at 40 °C in a water bath for 5 min, after which 1 mL of 6 mol/L HCl and 1 mL of diethyl ether were added, and then the mixture was centrifuged. Diethyl ether layer was collected in test tubes and evaporated under nitrogen gas to obtain 200 uL volume. A 200 uL aliquot was loaded onto a silica gel thin layer chromatography (TLC) plate with fluorescent indicator 254 nm and developed with hexane: diethyl ether acetic acid (50:50:1, v:v:v). 2-monoacylglycerol (2-MAG) band was visualised under UV light. 2-MAG band was scraped off into a screw capped test tube, extracted twice with 1 mL of diethyl ether, and centrifuged. The ether layer was collected and entirely evaporated under nitrogen, and then the sample was dissolved in n-hexane and methylated as described above.

### 2.6. Carotenoids Analysis

Breastmilk carotenoids’ (β-carotene, lycopene, and lutein + zeaxanthin) concentration in milk samples was assessed using high-performance liquid chromatography system (HPLC). Milk samples for the analysis were prepared on the basis of the modified method published earlier [12,13]. Analysis of the studied carotenoids was carried out at a wavelength of 471 nm for lycopene, 450 nm for β-carotene, and 445 nm for lutein + zeaxanthin using the HPLC system (Japan: 2 LC-20AD pumps, CMB-20A controller system, SIL-20AC autosampler, UV/VIS SPD-20AV detector, CTD-20AC controller, Shimadzu, Kioto, Japan) using C18 Synergi Fusion-RP 80i columns (250 × 4.60 mm, Phenomenex, CA, USA). The concentration of individual carotenoids was compared to standard curves prepared with Sigma Aldrich standards (catalogue numbers: β-carotene C4582, lutein + zeaxanthin X6250, lycopene L9879). The concentration of the studied carotenoids was expressed in nmol/L. Lutein and zeaxanthin could not be completely resolved and they were summed; therefore, all references to milk lutein concentration refer to lutein + zeaxanthin.

### 2.7. Statistical Analysis

Relative standard deviation for the lipase, lipidome, and lipolytic antioxidant factors results was calculated, where appropriate, for all collected data, using Microsoft Excel 2012 Software (Microsoft, Redmond, Washington, DC, USA). One-way analysis of variance (ANOVA) was performed using the Statgraphics Plus, version 5.1 (Statgraphics Technologies, Inc., The Plains, VA, USA). Differences were considered to be significant at a *p*-value ≤ 0.05, according to Tukey’s multiple range test.

## 3. Results

### 3.1. Lipase Activity

Because of the variability of lipase activity in human milk, the results from several experiments treating different milk samples were presented as a percent of activity in different samples with respect to raw milk considered 100%. As shown in Figure 1, the enzyme activity was nearly completely eliminated by HoP (2.1% residual activity). Higher lipase activity was detected in the HPP treated samples (16.5%—600 MPa, 11%—100 + 600 MPa, 13.6%—200 + 600 MPa). Almost entire lipase activity retention was observed in pressure 200 + 400 MPa, 10 min, interval 10 min; and 450 MPa, 15 min (82.2%—200 + 400 MPa, 87.3%—450 MPa). 

### 3.2. FFA Content

Any type of donor milk treatment was associated with a reduction in the FFA content compared with raw milk taken as 100% (Figure 2).

Holder pasteurization decreased FFA content by 50.2%. High pressure processing, depending on the variants, lowered FFA milk content by 19.5% for 100 + 600 MPa, 14.6% for 600 MPa, 12.8% for 200 + 400 MPa, 8.4% for 450 MPa, and 5.1% for 200 + 600 MPa (Figure 2).

### 3.3. Oxidative Stability

The PDSC measurements results of oxidative stability of sample expressed as the oxidation induction time (IT) are shown in Figure 3. Longer IT is connected with better stability. Significant differences in IT were observed especially between fat from raw and HoP milk—after heat treatment, IT was over 12 minutes shorter. The fat from samples after high pressure processing was also characterized by changed oxidative stability in comparison with the fat from raw human milk. The changes were slightly detectable in the case of 600 MPa and 200 + 600 MPa; more noticeable in 450 MPa, 15 min condition (3 minutes shorter); and about 7 min shorter induction time was observed in 200 + 400 MPa and 100 + 600 MPa (pressure 10 min, interval 10 min, pressure 10 min). 

### 3.4. Fatty Acid Composition

The fatty acids profile of fats from studied samples are presented in Figure 4 and Table 1. The fats from analyzed human milk contained from 10.7% to 12.2% of the polyunsaturated fatty acids (PUFA) and from 37.7% to 43.1% of the monounsaturated fatty acids (MUFA). Over 40% of all fatty acids in these fats were saturated fatty acids (SFA), of which the main representative was palmitic acid (from 23.8% to 25.1%). The main monounsaturated fatty acid present in analyzed samples of fats was oleic acid (from 34.0% to 39.4%). Polyunsaturated fatty acids found in studied samples were primarily linoleic acid (from 8.2% to 9.4%) and α-linolenic acid (1.3–1.4%). The major LCPUFA from n-6 family found in analyzed samples of fats was arachidonic acid (about 0.3–0.4%). Fats from studied human milk also contained omega-3 long chains polyunsaturated fatty acids like eicosapentaenoic acid (about 0.1%) and docosahexaenoic acid (about 0.3%). No statistically significant differences for samples after processing compared with raw milk fat were observed.

### 3.5. Fatty Acid Distribution

The results regarding the percentage of fatty acids esterified at sn-2 position of TAG from fats are presented in Figure 5 and Table 2. Palmitic acid was located at sn-2 position in 64.7% TAG from raw milk fats, 66.5% fats from milk after HoP. High pressure processing causes a slight increase in the palmitic acid in sn-2 position of TAGs—to about 70% (73.6% in 600 MPa, 72.6% in 200 + 400 MPa, 70.4% 200 + 600 MPa, and 72.8% for 450 MPa). The difference was not statistically significant. Oleic acid at sn-2 was detected in 15% of TAG extracted from raw and holder pasteurized milk samples. After HPP, the percentage of oleic acid at sn-2 position of TAG diminished to 12% for 600 MPa, as well as 200 + 400 MPa and 100 + 600 MPa. In the case of 200 + 600 MPa 10 min, interval 10 min, 10 min, the percentage of the fatty acids esterified at sn-2 position of TAG was 14.1%, and almost the same for 450 MPa, 15 min. (13%).

### 3.6. Carotenoids

The studied carotenoids’ concentration in raw milk was about 32.8 nmol/L for β-carotene, 85.9 nmol/L for lycopene, and 45.9 nmol/L for lutein + zeaxanthin. The content of carotenoids after processing compared with raw milk is presented in Figure 6.

β-carotene was intact after processing; therefore, the level was comparable in all milk samples. Statistically significant changes (*p* ≤ 0.05) were observed in the case of lutein and zeaxanthin, as well as lycopene concentration in milk samples after processing. Pasteurization diminished the content of lutein + zeaxanthin compared with raw milk to about 84.1%, whereas HPP was so destructive as to leave 39.7% after 600 MPa, 52.9% after 200 + 400 MPa, 42.5% after 200 + 600 MPa, 60% after 100 + 600 MPa, and 42.4% after 450 MPa. In the case of lycopene, a slight content increase was observed after processing compared with raw milk—8.8% in case of pasteurization; 5.7% after 450 MPa; and 111.8% to 113.6% after other variants of high pressure processing, 200 + 400 MPa and 200 + 600 MPa, respectively. 

## 4. Discussion

The main practical advantages of putting the baby to the breast for feeding are overall integrity of given nutrients, optimal temperature, microbiological safety, and intact bioactivity of mother’s milk. Every other way of feeding involves some losses in human milk properties. Human milk is a complex body fluid with thousands of components in dynamic interplay [14,15,16]. For example, human milk fat breakdown by endogenous lipase is rather not noticeable when the baby is fed directly from the breast. In contrast, expressing human milk with a high lipase activity is associated with free fatty acids release from TAG, even if milk is frozen immediately. The increase of FFA levels causes pH changes that affect enzyme activity and other inherent human milk components [17,18,19]. In addition, unbound FFA may cause cytotoxicity in the intestinal lumen [20]. However, endogenous lipases are the prime effector of milk fats’ digestion, facilitating human milk consumption in the early period of life. Therefore, it is crucial to retain enzymatic properties of human milk, especially when human milk is banked for preterms. As several previous studies proved, the activity of BSSL in donor human milk is mostly suppressed by holder pasteurization [21,22,23]. In contrast, high pressure processing seems to be a favorable technique for human milk preservation for this purpose [24]. Our current results concerning the residual enzymatic activity of lipase after HPP are consistent with the discovery of Demazou and co-workers, as well as the work of Pitino et al. [25,26]. All studies showed slightly diminished functional activity of this enzyme in milk after processing it in selected pressure conditions to about 80%, compared with 100% in raw milk [26]. In fact, the difference between detected enzyme activity in raw milk and milk treated in pressure of 450 MPa for 15 min and 200 + 400 MPa for 10 min with a 10 min interval was not statistically significant (Figure 1).

As human milk enzymes such as BSSL modulate the digestion to favor absorption of triglycerides hydrolysis and vitamins, human milk processing may negatively affect infants’ ability to digest lipids because of the total inactivation of endogenous human milk lipases [27].

Human milk FFA content seems to be a good indicator of BSSL bioavailability in processed milk, because in the case of human milk, the FFA concentration should depend mostly on this enzyme’s activity. However, available data concerning the effect of HoP on FFA concentration were inconsistent. Lepri et al. reported an 83% increase in FFA content after heat treatment [28], in contrast to several studies where no change was detected [29,30,31,32]. Only Williamson et al. report a 21% decrease in FFA concentration [33]. As shown in this study, all applied preservation techniques reduced the content of free fatty acids in fat extracted from human milk with a statistically significant difference (Figure 2). In the case of pasteurized milk, a reduction as large as 50% in the content of FFA compared with raw milk was noticed. FFA content decrease in fat from high pressure processed human milk was from 5% to 19% compared with fat from raw human milk, which is consistent with previous studies [30]. 

Analyzing the results, it can be observed that the differences in FFA fat content are especially significant in milk pasteurized by HoP, while after high pressure processing, the decrease in FFA content is minor. So when the enzyme was completely destroyed by heat treatment in the HoP samples, the FFA content was the lowest. This rule does not apply to high pressure samples. Among HPP variants, the largest reduction in FFA content was noticed in milk subjected to 100 + 600 MPa pressure, and lipase activity in this variant is significantly diminished. In contrast to this finding, at 200 + 600 MPa, the FFA content is almost intact while enzyme activity is severely lowered (Figure 1, Figure 2). In fact, our results of FFA and lipase activity in human milk after processing are ambiguous. The reason for this discrepancy could be a very low level of FFA in untreated human milk taken as 100%—as a result, any changes given as a percentage lead to large proportional differences [34]. 

Nevertheless, based on our research, we concluded that the best option seems to be 450 MPa for 15 min as the most favorable conditions for preservation of FFA concentration and lipase activity.

Oxidative stability is one of the most important factors determining the quality of foods, especially those containing high levels of unsaturated fats, including human milk. Within the fatty acid family, polyunsaturated fatty acids are highly labile molecules susceptible to oxidation, giving rise to free radicals, hydroperoxides and polymers, which might lead to loss of quality, both technological and related to their health benefits [35]. A number of methods for the assessment of oxidative stability have been developed, among which differential scanning calorimetry is one of the most frequently used. This method is very fast and convenient [36,37,38]. Generally, samples with longer induction times (IT) are more stable than those with a shorter induction time at the same temperature [37,39].

In our study, IT changes compared with raw milk value ranged from 5% for samples from fat after HPP in 600 MPa to 46% for HPP variant with 200–400 MPa. It appears that any kind of preservation causes changes to human milk oxidative status, but less so in selected pressure variants than in HoP (Figure 3).

Although original causes and consequences of oxidative and hydrolytic degradation processes are quite different, they seem to interact with each other and contribute to the reduction in stability of fats. Research suggests that fat FFA presence may induce oxidation as a result of a catalytic effect of carboxylic groups of FFA on the formation of free radicals. In general, the higher the level of fatty acids content, monoacylglycerols, and diacylglycerols in the fat with respect to the level of TAG, the more the oxidative stability is reduced. Pro-oxidant FFA action is thus connected with carboxylic molecular group, which accelerates the rate of hydroperoxides decomposition [35,40,41].

Analyzing the results obtained in this work, it can be concluded that the reduction in FFA content did not affect the improvement of oxidative fat stability (Figure 2, Figure 3). Pasteurized milk, despite the low content of FFA, is characterized by the shortest induction time, and thus the worst oxidative stability. Deterioration of oxidative stability in the case of pasteurized human milk could have been influenced by other factors, such as temperature, that could affect the content of antioxidants in the samples.

Human milk contains 3–5% of lipids and approximately 99% of them are TAG. The major fatty acids of TAG are palmitic acid (16:0), oleic acid (18:1), and linoleic acid (18:2 *n*-6), which constitute 25%, 30%, and 15%, respectively, of all fatty acids [42,43]. Therefore, fatty acid composition of human milk fat is unique, because the fat is characterized by a high content of saturated palmitic acid and also contains polyunsaturated fatty acids, which are not present in other milk fats [44,45,46]. 

Polyunsaturated fatty acid components of human milk are complex, including both C18 precursors, linoleic acid (18:2 *n*-6) and α-linolenic acid (18:3 *n*-3), as well as bioactive, very long chain polyunsaturated fatty acids (LCPUFA) of both *n*-6 and *n*-3 families [46,47]. 

Most of the studies on the effect of human milk processing, including the current study, have claimed that fatty acid composition was intact after holder pasteurization [21,30,47]. In addition, the results from our study proved no change in fatty acid profile in human milk after HPP, which is consistent with discoveries from other research studies on the effect of HPP on human milk [21,30]. It can be concluded that high pressure processing and holder pasteurization do not affect the composition of fatty acid (Figure 4, Table 1). 

Moreover, the structure of human milk TAG is also unique, as 60–70% of palmitic acid is located at sn-2 position, and sn-1 and sn-3 positions are taken by 18:0, 18:1, and 18:2 fatty acids. This unique intramolecular structure is one of the key factors controlling the products formed by the gastric lipase in the stomach and by pancreatic or bile salt-stimulated lipases in the small intestinal absorption; therefore, it improves the efficiency of calcium absorption [48]. 

Considering the composition of fatty acid in sn-2 position of monoacylglycerols from the studied fats (Table 2), it can be seen that saturated fatty acids that occur mainly in this position are palmitic and myristic acid and unsaturated fatty acids—oleic and linoleic acids.

Analyzing the results regarding the percentage of fatty acids esterified at sn-2 position of TAG from the studied fats, it can be concluded that differences in the distribution of fatty acid in TAG were really small (Figure 5). There are no statistically significant differences between raw milk and HoP in the distribution of fatty acid in triacylglycerol molecules. These results are confirmed by many research studies [5,29,30,31,47,49]. According to scientists, no difference was found in the bioaccessibility of different fatty acids released from raw human milk and pasteurized human milk. If pasteurization affected the structure of triacylglycerols (TAG), the bioavailability of fatty acids found in these TAG would also change.

The percentage of palmitic acid and myristic acid at sn-2 of TAG in fats from all processed samples of human milk exceeded 33% (ranged from 64.7% to 73.6% for palmitic acid and from 46.7% to 61.4% for myristic acid), which means that these saturated fatty acids are located mainly in the internal position of TAG (Figure 5). This location increases the efficiency of calcium absorption in infants. Taking into account the percentage of unsaturated fatty acids in sn-2 position of TAG in studied fats, it can be stated that they are located mostly in external positions of TAG. The percentage of oleic acid at sn-2 of TAG in analyzed fats ranged from 12.2% to 15.7% and linoleic acids—from 19.2% to 26.3%, which confirms that they are located mainly in sn-1,3 positions of TAG (external position of TAG). 

Carotenoids are principal non-enzymatic, lipophilic antioxidants that act as free radical scavengers. There is a substantial and growing body of research evidence showing an important role of carotenoids, especially lutein, in human development, related to their antioxidant and anti-inflammatory properties [50,51]. This is crucial for preterm infants, for whom carotenoid (mainly lutein) administration may decrease the risk of several prematurity disorders, including necrotizing enterocolitis, bronchopulmonary dysplasia, and retinopathy of prematurity related to elevated oxidative stress [52,53,54]. Carotenoids are one of the bioactive factors in human milk, and their amount is determined by maternal dietary intake; infant formulas are not fortified enough [51,55]. In our study, we found that all variants of HPP slightly changed the β-carotene and lycopene content (it is increased), whereas lutein + zeaxanthin content is decreased by 40.0% (at 600 MPa, 200 + 600 MPa, 450 MPa) up to 60.2% (at 100 + 600 MPa) compared with a 15.8% decrease in the holder pasteurized samples. It is the first time that the effect of high pressure on human milk with regard to β-carotene content has been evaluated. The only existing study comparing maternal breastmilk with donor milk has revealed the 18–53% lower amount of carotenoids after heat treatment, which is still higher than in infant formulas [55]. The increase in lycopene concentration after high pressure treatment was observed before in tomato products [56]. Human milk preservation and storage are not the only processes that affect carotenoid content. A study conducted by Tacken et al. also found that lutein is more sensitive to processing than other carotenoids [57]. In this case, tube feeding was related to the decreased concentration of lutein by 35% (even 42.6% with the phototherapy exposure), β-carotene by 26%. This relatively high decrease in lutein content may be caused by its high susceptibility to oxidation damage.

## 5. Conclusions

Human milk lipids are essential nutrition components for the growth and development of an infant. Commonly used holder pasteurization affects some of the nutritional and biological components of human milk and its lipid fractions [21,34]. Moreover, structural disintegration of lipid fraction of human milk by thermal pasteurization has been proven [7,27]. However, in the pilot clinical trial, the significance of those changes has not been proven with regards to important nutritional outcomes such as effectiveness of gastric emptying in the group of preterm infants [58,59]. Other studies revealed a decrease of fat absorption in preterms fed with donor milk [60]. We determined that high pressure processing, especially at 450 MPa for 15 min., minimizes changes in the lipidome and lipid related components such as lipase of donor human milk, with the exception of the destructive effect on lutein + zeaxanthin. As was previously shown, preterms’ growth rate during their hospital stay depends on overall quality of lipidome [61]. Therefore, any methods of improving human milk lipids preservation, including high pressure processing of donor milk, could be beneficial in managing optimal infant weight gain and growth.

## Figures and Tables

**Figure 1 nutrients-11-01972-f001:**
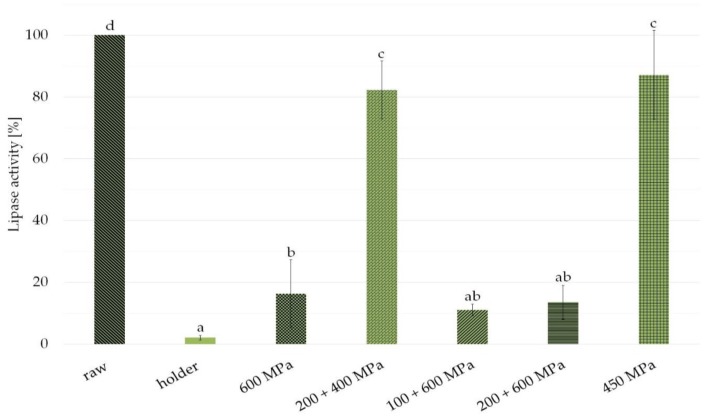
Lipase enzymatic activity in milk samples after holder pasteurization and variants of high pressure processing (HPP) compared with raw milk. Results are shown as a % value of raw milk with error bars representing SD. Different letters indicate that the samples are significantly different at *p*-value < 0.05.

**Figure 2 nutrients-11-01972-f002:**
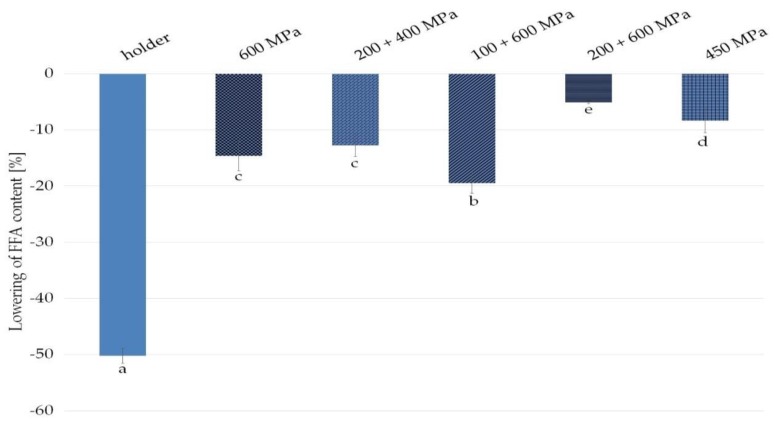
Decrease in free fatty acids (FFA) content compared with raw milk in fats from samples of milk after HPP and holder pasteurization. Results are shown as a % value of raw milk with error bars representing SD. Different letters indicate that the samples are significantly different at *p* < 0.05.

**Figure 3 nutrients-11-01972-f003:**
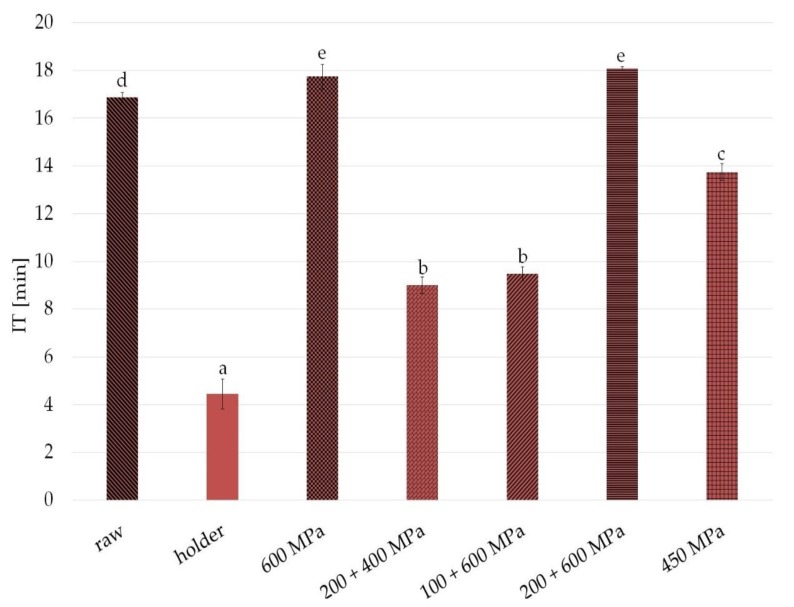
Oxidative stability of raw milk, in the samples after HPP and holder pasteurization. Results are shown as a % value of raw milk with error bars representing SD. Different letters indicate that the samples are significantly different at *p* < 0.05. IT, induction time.

**Figure 4 nutrients-11-01972-f004:**
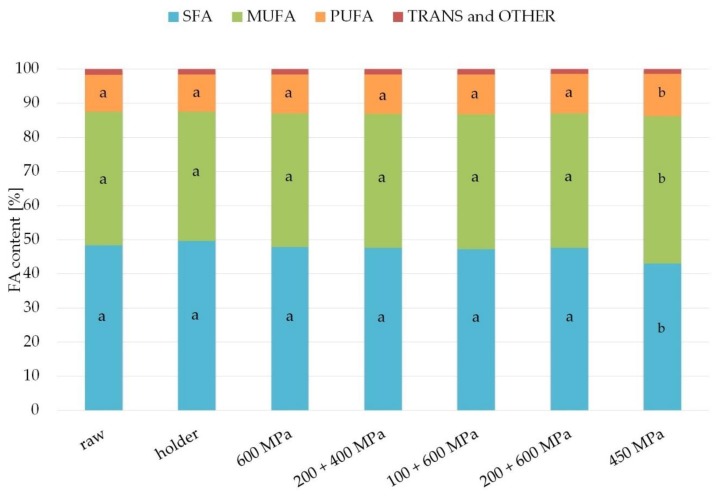
Content of fatty acid (SFA, saturated fatty acids; MUFA, monounsaturated fatty acids; PUFA, polyunsaturated fatty acids; TRANS, trans fatty acids) in the samples after HPP and holder pasteurization. The different lower case letters (for SFA, MUFA, and PUFA separately) indicate significantly different values (*p* < 0.05).

**Figure 5 nutrients-11-01972-f005:**
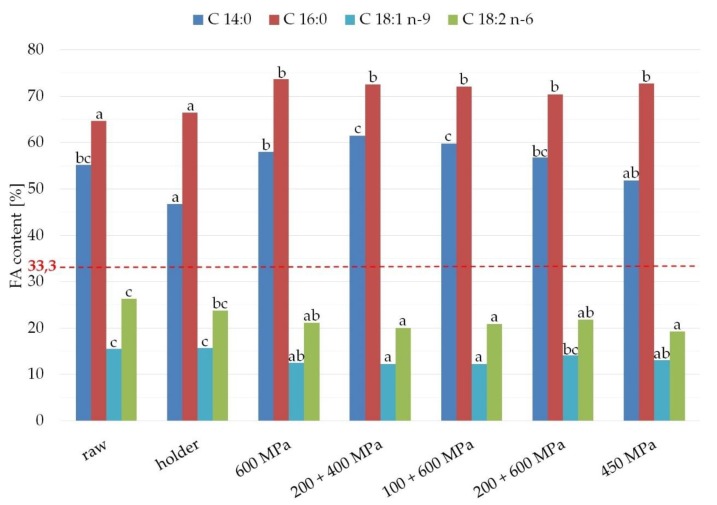
Fatty acid distribution in the sn-2 position of TAG from raw milk and in the samples after HPP and holder pasteurization. The different lower case letters for each fatty acid separately indicate significantly different values (*p* < 0.05). The line indicates the statistical (even) distribution of fatty acids between three TAG positions (33%).

**Figure 6 nutrients-11-01972-f006:**
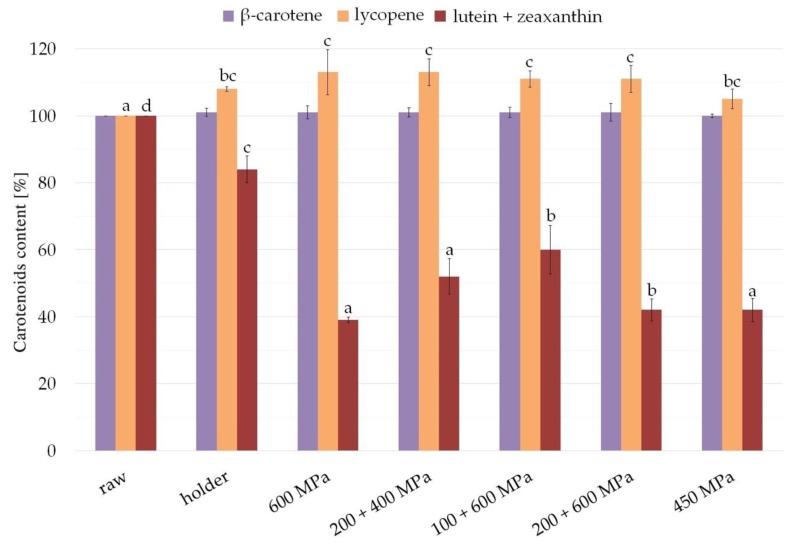
Content of carotenoids (β-carotene, lycopene, and lutein + zeaxanthin) in milk samples after HPP and holder pasteurization. Results are shown as a % value of raw milk with error bars representing SD. Different letters indicate that the samples are significantly different at *p* < 0.05.

**Table 1 nutrients-11-01972-t001:** Changes in fatty acid composition during high pressure processing (HPP) and holder pasteurization (mean ± SD).

Human Milk Fatty Acid Composition (%)	Raw Milk	Holder	600 MPa	200 + 400 MPa	100 + 600 MPa	200 + 600 MPa	450 MPa
**Saturated fatty acids (SFA)**	48.53 ± 0.94	49.77 ± 3.39	47.94 ± 0.83	47.70 ± 0.25	47.32 ± 0.08	47.72 ± 0.18	43.17 ± 0.62
**C8:0**	0.06 ± 0.04	0.11 ± 0.02	0.08 ± 0.04	0.04 ± 0.01	0.06 ± 0.01	0.05 ± 0.01	0.05 ± 0.00
**C10:0**	0.89 ± 0.09	1.07 ± 0.23	0.83 ± 0.10	0.77 ± 0.01	0.88 ± 0.02	0.85 ± 0.02	0.73 ± 0.09
**C12:0**	5.77 ± 0.14	6.58 ± 1.26	5.72 ± 0.36	5.28 ± 0.05	5.76 ± 0.11	5.63 ± 0.08	3.96 ± 0.51
**C14:0**	8.47 ± 0.40	8.89 ± 1.48	8.27 ± 0.35	8.08 ± 0.13	8.10 ± 0.07	8.15 ± 0.08	6.11 ± 0.39
**C15:0**	0.36 ± 0.02	0.38 ± 0.05	0.35 ± 0.02	0.35 ± 0.01	0.35 ± 0.01	0.35 ± 0.00	0.36 ± 0.01
**C16:0**	24.59 ± 0.49	25.13 ± 1.30	24.22 ± 0.08	24.45 ± 0.23	23.80 ± 0.01	24.14 ± 0.02	23.93 ± 0.16
**C17:0**	0.36 ± 0.03	0.33 ± 0.00	0.33 ± 0.01	0.34 ± 0.01	0.33 ± 0.01	0.34 ± 0.00	0.34 ± 0.01
**C18:0**	7.57 ± 0.03	6.94 ± 0.83	7.73 ± 0.05	7.95 ± 0.14	7.64 ± 0.09	7.81 ± 0.01	7.26 ± 0.20
**C20:0**	0.47 ± 0.03	0.36 ± 0.11	0.43 ± 0.01	0.46 ± 0.01	0.42 ± 0.01	0.43 ± 0.01	0.45 ± 0.01
**Polyunsaturated fatty acids (PUFA)**	39.07 ± 0.15	37.715 ± 2.34	39.04 ± 0.70	39.24 ± 0.42	39.46 ± 0.15	39.25 ± 0.08	43.13 ± 0.45
**C20:1**	0.52 ± 0.01	0.43 ± 0.13	0.52 ± 0.01	0.55 ± 0.04	0.50 ± 0.01	0.52 ± 0.01	0.72 ± 0.01
**C14:1**	0.25 ± 0.04	0.32 ± 0.04	0.29 ± 0.00	0.27 ± 0.01	0.28 ± 0.00	0.28 ± 0.00	0.23 ± 0.01
**C15:1**	0.10 ± 001	0.11 ± 0.02	0.09 ± 0.00	0.10 ± 0.01	0.09 ± 0.01	0.09 ± 0.00	0.08 ± 0.00
**C16:1**	2.63 ± 0.35	2.69 ± 0.25	2.44 ± 0.04	2.45 ± 0.11	2.46 ± 0.01	2.46 ± 0.01	2.49 ± 0.02
**C17:1**	0.20 ± 0.01	0.21 ± 0.03	0.21 ± 0.01	0.20 ± 0.01	0.23 ± 0.01	0.22 ± 0.01	0.20 ± 0.00
**C18:1**	35.39 ± 0.46	33.97 ± 2.50	35.49 ± 0.74	35.68 ± 0.25	35.91 ± 0.13	35.69 ± 0.09	39.41 ± 0.47
**Polyunsaturated fatty acids (PUFA)**	10.73 ± 1.00	11.04 ± 1.05	11.4 ± 0.12	11.53 ± 0.08	11.67 ± 0.07	11.57 ± 0.00	12.24 ± 0.30
**C18:2 n-6**	8.17 ± 1.15	8.75 ± 0.64	8.90 ± 0.06	8.99 ± 0.01	9.08 ± 0.04	9.05 ± 0.01	9.40 ± 0.18
**C18:3 n-3**	1.36 ± 0.12	1.28 ± 0.12	1.27 ± 0.01	1.30 ± 0.03	1.36 ± 0.05	1.30 ± 0.00	1.33 ± 0.06
**C20:2 n-6**	0.20 ± 0.01	0.18 ± 0.04	0.22 ± 0.01	0.22 ± 0.01	0.22 ± 0.01	0.22 ± 0.00	0.32 ± 0.00
**C20:3 n-6**	0.20 ± 0.01	0.21 ± 0.07	0.27 ± 0.03	0.25 ± 0.01	0.26 ± 0.00	0.25 ± 0.00	0.35 ± 0.01
**C20:4 n-6**	0.38 ± 0.01	0.34 ± 0.08	0.38 ± 0.01	0.39 ± 0.00	0.39 ± 0.00	0.38 ± 0.00	0.44 ± 0.01
**C20:5 n-3**	0.14 ± 0.01	0.07 ± 0.01	0.11 ± 0.04	0.11 ± 0.02	0.09 ± 0.01	0.10 ± 0.00	0.09 ± 0.03
**C22:6 n-3**	0.31 ± 0.01	0.22 ± 0.08	0.28 ± 001	0.27 ± 0.03	0.29 ± 0.01	0.28 ± 0.01	0.31 ± 0.00

**Table 2 nutrients-11-01972-t002:** Changes in the fatty acid composition of sn-2 monoacylglycerols during HPP and holder pasteurization (mean ± SD; only selected fatty acids are presented).

Fatty Acid (%) in sn-2 Position of TAG	Raw Milk	Holder	600 MPa	200 + 400 MPa	100 + 600 MPa	200 + 600 MPa	450 MPa
**C14:0** **(myristic acid)**	14.00 ± 0.71	12.50 ± 0.50	14.40 ± 0.47	14.90 ± 1.12	14.52 ± 0.62	13.90 ± 0.63	9.50 ± 0.54
**C16:0** **(palmitic acid)**	47.70 ± 0.81	50.10 ± 1.34	53.50 ± 1.85	53.20 ± 0.59	51.50 ± 0.45	51.00 ± 1.22	52.20 ± 0.02
**C18:1** **(oleic acid)**	16.40 ± 0.35	16.00 ± 0.18	13.40 ± 0.04	13.10 ± 1.07	13.20 ± 0.07	15.10 ± 1.65	15.40 ± 0.25
**C18:2** **(linoleic acid)**	6.50 ± 0.28	6.20 ± 0.10	5.70 ± 0.11	5.40 ± 0.57	5.67 ± 0.28	5.90 ± 0.28	5.40 ± 0.28

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
