# Peer review of "Lipid Profile, Lipase Bioactivity, and Lipophilic Antioxidant Content in High Pressure Processed Donor Human Milk"

_nutrients, 2019, doi:10.3390/nu11091972_

Round 1
Reviewer 1 Report
The aim of the paper is to compare the effects of pascalization and holder pasteurization on lipidome. Lipase enzymatic activity and oxidative stability are preserved after certain variants of pascalization procedures. Although pascalization decreased content of lutein+zeaxanthin more than holder pasteurization.
The effect on lipase activity may favour lipid digestibility and growth. Oxidative stability may decrease oxidative derivatives of LCPUFA. Digestivility and proof of less oxidative products can improve the significance of the recommendation of pascalization over Holder pasteurization.
Figure 4 & 5 and Table 1 & 2 should be improved. Page 14, line 395: "Human milk contains 3-5% of lipids", explain
Reviewer 2 Report
This paper describes a useful study of the relative effects of high pressure processing as an alternative to high temperature processing of human milk. The experiments appear to be well designed, executed, and reported. There are numerous issues with the presentation that require attention. Whomever of the authors was responsible for presentation of the basic facts of the projects should take on the rest of the presentation to put it in good shape.
The first striking thing is the abstract that is full of undefined abbreviations and jargon and is generally not easy to follow. I'd start again with a blank sheet, use no abbreviations, and concisely describe the study.
Below are mixed major and minor points.
l55 What are "potential losses in human milk lipidome"?
l57 "...loss of milk fatty acid composition..." This phrase has no discernable meaning.
l65. The term "pascalization" is not familiar to most nutritionists, nurses, pediatricians or others likely to be interested in this topic, and should be defined in a paragraph prior to this final paragraph of the Introduction.
l82. the term "high pressure processed" is introduced here but not defined as an abbreviation as used in the abstract. Confusing.
l87 "Lipase activity assay..." (assays?)
l164 "Carotenoid concentrations..."
Figures. Many axes on figures have no label. Every axis must be labeled with the parameter and the units, e.g. Weight (kg).
l243 "Fatty acids composition" should be "Fatty acid composition"
l272 Same comment. The proper English is "Fatty acid" not the plural.
l296 proper spelling is "monoacylglycerols"
l323. what could be meant by the assertion that breakdown of human milk fat by endogenous lipase is not rather noticeable? See many papers by Hamosh dated 1970s-2000s on lingual lipase and hydrolysis in the stomach of 30% within 10 minutes.
The Discussion is far too long and most of the discussion of biology (DHA meta-analyses and bias?) is irrelevant. It should be cut back to half or less.
Round 2
Reviewer 2 Report
The manuscript is much improved over the original submission. However the abstract remains a problem. Several issues remain.
l26 "...pascalization (HPP)..."
Abbreviations are usually chosen to be suggestive of the full word. Here we have an abbreviation for "high pressure processing" that is used for pascalization. I renew my opinion that likely readers do not know what pascalization is, and see no advantage in using this term rather than the also-standard HPP which is much more obvious to those new to the topic.
l28. "Lipase (BSSL)...", again an abbreviation which is not obvious from the sentence.
l28 "Fatty acids (FA) composition..."
A simple search would reveal all instances of this error which was said to be corrected. Please perform it.
l28. "Fatty acids (FA) composition was determined with GC technique, free FA content by titration with 0.1M KOOH." this sentence has numerous errors of grammar and fact, at least one pointed out in the original review but missed. What is KOOH?
l31. PDSC. Undefined abbreviation.
Author Response
Review #2 round 2
Reply: We thank the reviewer for their careful reading of the manuscript and their constructive remarks. We have taken the comments on board to improve and clarify the manuscript. In the first version, we changed the content of the abstract. We don’t know why the old version is still visible on the MDPI page.
Please find below a detailed point-by-point response to all comments (our replies in Italic).
As suggested by Reviewer #2, we have changed the abstract section. We believe that this makes the manuscript substantially clearer.
Comments and Suggestions for Authors
The manuscript is much improved over the original submission. However the abstract remains a problem. Several issues remain.
l26 "...pascalization (HPP)..."
Abbreviations are usually chosen to be suggestive of the full word. Here we have an abbreviation for "high pressure processing" that is used for pascalization. I renew my opinion that likely readers do not know what pascalization is, and see no advantage in using this term rather than the also-standard HPP which is much more obvious to those new to the topic.
Reply: We agree that this wording could be unclear. When we use the pascalization, this means that we are referring to the HPP. We have amended this, now in the manuscript, we have high pressure processing or HPP.
l28. "Lipase (BSSL)...", again an abbreviation which is not obvious from the sentence.
Reply: We have revised the abstract to clarify the abbreviation.
l28 "Fatty acids (FA) composition..."
A simple search would reveal all instances of this error which was said to be corrected. Please perform it.
l28. "Fatty acids (FA) composition was determined with GC technique, free FA content by titration with 0.1M KOOH." this sentence has numerous errors of grammar and fact, at least one pointed out in the original review but missed. What is KOOH?
Reply: We apologies for any confusion. The abstract has been improved, language errors have been corrected.
l31. PDSC. Undefined abbreviation.
Reply: We have revised the abstract to clarify the abbreviation.
We would like to thank the reviewer for all comments that will make this paper more valuable for readers. The present form of the manuscript has been modified according to the reviewer‘s suggestions. We take your concerns seriously and have addressed them to the best of our abilities (we've added a new abstract below).
Abstract: Human milk fat plays an essential role as the source of energy and cell function regulator, therefore the preservation of unique human milk donor's lipid composition is of fundamental importance. To compare the effects of high pressure processing (HPP) and holder pasteurization on lipidome, human milk was processed at 62.5°C for 30 min and at five variants of HPP from 450 MPa to 600 MPa, respectively. Lipase activity was estimated with QuantiChrom™ assay. Fatty acid composition was determined with gas chromatographic technique, free fatty acids content by titration with 0.1M KOH. The positional distribution of fatty acid in triacylglycerols was performed. The oxidative induction time was obtained from the pressure differential scanning calorimetry. Carotenoids in human milk were measured by liquid chromatography. Bile salt stimulated lipase was completely eliminated by holder pasteurization, decreased at 600MPa and remained intact at 200+400MPa; 450MPa. Fatty acid composition and structure of human milk fat triacylglycerols were unchanged. The lipids of human milk after holder pasteurization had the lowest content of free fatty acids and the shortest induction time compared to samples after HPP. HPP slightly changed theβ-carotene and lycopene levels, whereas lutein level was decreased by 40.0% up to 60.2%, compared to 15.8% after the holder pasteurization.